# A Relativistic Entropic Hamiltonian–Lagrangian Approach to the Entropy Production of Spiral Galaxies in Hyperbolic Spacetime

Michael C. Parker [1] and Chris Jeynes [2,*]

1 School of Computer Science and Electrical Engineering, University of Essex, Wivenhoe Park, Colchester CO4 3SQ, UK; mcpark@essex.ac.uk
2 University of Surrey Ion Beam Centre, Guildford GU2 7XH, UK
* Correspondence: c.jeynes@surrey.ac.uk

**Abstract:** Double-spiral galaxies are common in the Universe. It is known that the logarithmic double spiral is a Maximum Entropy geometry in hyperbolic (flat) spacetime that well represents an idealised spiral galaxy, with its central supermassive black hole (SMBH) entropy accounting for key galactic structural features including the stability and the double-armed geometry. Over time the central black hole must accrete mass, with the overall galactic entropy increasing: the galaxy is not at equilibrium. From the associated entropic Euler–Lagrange Equation (enabling the application of Noether's theorem) we develop analytic expressions for the galactic entropy production of an idealised spiral galaxy showing that it is a conserved quantity, and we also derive an appropriate expression for its relativistic entropic Hamiltonian. We generalise Onsager's celebrated expression for entropy production and demonstrate that galactic entropy production (entropy production corresponds to the intrinsic dissipation characteristics) is composed of two parts, one many orders of magnitude larger than the other: the smaller is comparable to the Hawking radiation of the central SMBH, while the other is comparable to the high entropy processes occurring within the accretion disks of real SMBHs. We conclude that galaxies cannot be isolated, since even idealised spiral galaxies intrinsically have a non-zero entropy production.

**Keywords:** Quantitative Geometrical Thermodynamics; *info-entropy*; galactic evolution; *Maximum Entropy Production Principle*; accretion

## 1. Introduction

The logarithmic double-spiral is ubiquitous in Nature, being seen for example in sunflower seed patterns, the shells of Nautilus snails, cyclones, and spiral galaxies. Parker and Jeynes [1] ("**PJ19**") have demonstrated that it has a Maximum Entropy (**MaxEnt**) geometry, in the framework of their theory of Quantitative Geometrical Thermodynamics (**QGT**). The idea of a geometry having an entropy is counter-intuitive but is a consequence of the systematic consideration of the complementary nature of information and entropy that has become accepted since Shannon's seminal introduction of his "*information entropy*" [2]. We should note that the concept of a "geometric entropy" is already well accepted in quantum gravity contexts (see, for example, Vacaru et al. [3]) but is not usually treated as an intrinsic property of geometrical structures, although perhaps Quevedo's "*Geometrothermodynamics*" [4] is an exception. We should also note not only that Wang et al. [5] have recently specifically investigated the *entropy production* of certain natural growth processes but also that Pearson et al. [6] have recently shown that a chronometer's *accuracy* is proportional to its *entropy production*. On "accuracy", Parker and Jeynes [7] ("**PJ21**") have also recently established the *entropic* Uncertainty Principle.

Here we rely on our previous fundamental work (PJ19, [1]) demonstrating a coherent and comprehensive *entropic* isomorph to the *Principle of Least Action* (**PLA**): the *Principle of*

*Least Exertion* (**PLE**). The PLA is based upon Planck's constant $h$ (with units of [Js]), and is understood to underpin much of 'known' physics (see Coopersmith's review [8]: in QGT the reduced Planck constant $\hbar$ is isomorphic to $2ik_B$, where $i^2 = -1$ and $k_B$ is the Boltzmann constant with units of [J/K]; see Eq.15 of PJ21, [7]). The PLA considers the trajectory of a system's energy over time, whereas the PLE is entropic in origin (being fundamentally based upon $k_B$) and describes the spatial 'trajectory' (that is, the geometry) of a system and its entropy over space. Thus, the PLE can be understood to underpin the concept of geometric entropy.

In PJ19 we proved the mathematical properties of the PLE, demonstrating how it obeys the entropic Euler–Lagrange Equations (calculus of variations), with Noether's theorem indicating the existence of various entropic conservation laws. PJ19 also proved that the ubiquity of double logarithmic spirals in the natural world (as seen in spiral galaxies such as the Milky Way) is based upon a universal physical principle: the double-armed structure of a spiral galaxy (featuring at least one *pair* of arms) conforms to the PLE and represents a *maximum entropy* (MaxEnt) geometry. More recently, PJ21 have also proved the *entropic* version of Liouville's theorem, in particular how it relates to the conservation of PLE trajectories in an *entropic* phase space based upon Boltzmann's constant $k_B$. This describes the mapping of a MaxEnt geometric structure into an incompressible (and therefore conservative) *entropic* phase space. Together, these fundamental results in the analytical properties of geometric entropy indicate the existence of profound conservation laws operating within the entropic domain; entirely isomorphic to the more familiar energetic conservation laws based upon the PLA.

The concept of "entropic conservation laws" might appear paradoxical: issues of dissipation (due to the effects of friction and viscosity etc.) are conventionally considered to preclude conservation (although, of course, the 1st Law remains inviolate): "conservation" is thought to be reserved for thermodynamically reversible systems. However, the PLE in combination with Noether's and Liouville's Theorems demonstrates that the entropic domain also features conservation laws across spacetime, just as rigorous and far-reaching as those exhibited by systems viewed from the perspective of energy.

That is to say, the entropic domain is entirely isomorphic to the better-known energetic domain; indeed, whereas issues of dissipation and friction ("*entropy production*") are frequently considered annoying imperfections of the 'real' world, QGT indicates that entropy production is as primary in any model of a real system as is its energy. The entropic domain provides an analytic description of phenomena in spacetime that is as mathematically elegant as their descriptions in the energy domain (with all its associated fundamental symmetries and variational principles).

In this paper, we build upon the foundational work of PJ19 and PJ21 to now demonstrate a coherent and *relativistically invariant* description of system dynamics in the entropic domain, isomorphic to the application of Special Relativity principles in the energy domain. Entropic considerations shed new light onto the intimate relationship between the geometric structure of a spiral galaxy and the dissipative processes intrinsic to it.

We also continue to build on the description of spiral galaxies as MaxEnt unitary objects obeying the PLE, but now also featuring fundamental entropy production characteristics intrinsic to their geometric structure. That is to say, entropy production (the rate of entropy increase over time) is a fundamental and intrinsic feature of spiral galaxies and critically dependent on their geometric properties. Thus, we find that the entropic domain is entirely complementary to the energetic domain: for any system, either the energetics or the entropic behaviour (or both) can be studied. Both perspectives are necessarily consistent with each other, yet each yields complementary insights into the overall characteristics of the system. We believe that both are needed.

The stability of a geometry implies a time invariance, and if it is also in static equilibrium then no entropy is being produced. PJ19 considered "information" specifically as a 3-D integration over time in Minkowski 4-space and showed that *entropy* and *information* are mutually Hodge-duals, where *info-entropy* can have the same geometry as the

electromagnetic field in free-space (the double-helix). Both photons and DNA have a double-helix geometry, which PJ19 proved MaxEnt, and both photons and DNA are very stable. PJ19 (their Figure 1 and context) demonstrated that the weird assertion of DNA having a "geometrical entropy" is actually supported in detail by observation.

Now the double-helix (instanced by DNA and photons) is a stable geometry having zero entropy production (being in static equilibrium). The logarithmic double-spiral (of which the double-helix is only a special case) is also MaxEnt, but being instanced by living systems (sunflowers, Nautiluses) and violent systems (cyclones, galaxies) demonstrates that being MaxEnt still permits a *dynamic* thermodynamic equilibrium: entropy is certainly being produced in these systems! In this work we develop QGT to display the far-reaching implications of the theory in considering the evolution in time of such MaxEnt systems.

In particular, PJ19 explicitly showed that spiral galaxies could be idealised as logarithmic double-spirals in a full *entropic* Lagrangian–Hamiltonian treatment (see their Figure 2 and context). Again, this representation is supported in detail by observation: "the galactic *shape*, *aspect ratio* and *structural stability* (which are all highly constrained by the algebra) are consistent with observation" (PJ19 p8: italics original); moreover, an eigenvalue of the appropriate entropic Hamiltonian can be related to the galactic *virial mass*, also taking a realistic value.

A few further remarks are needed to set the scene. Variational principles have a fundamental importance in physics: the *Principle of Least Action* also applies (*mutatis mutandis*) in quantum mechanics, as Richard Feynman famously pointed out in 1942 [9]. Edwin Jaynes first formulated the principles of *Maximum Entropy* [10] and *Maximum Caliber* [11] (helpfully reviewed by Pressé et al. [12] and further expounded by Dixit et al. [13]), and of course we here follow Willard Gibbs [14,15] and Lars Onsager [16] who both, saw the central importance of the variational calculus in the entropic context. Hans Ziegler saw the implications of Onsager's work in his *Maximum Entropy Production Principle* [17] which he subsequently developed [18], while Martyushev and Seleznev [19] explain that Ilya Prigogine's apparently contradictory *Minimum Entropy Production Principle* [20] is actually complementary. A helpful context to entropy production is also offered by Walter Grandy in his 2008 textbook [21].

Entropy production is closely related to the fluctuation theorem (Evans and Searles, 1994 [22]): this and the *Maximum (or Minimum) Entropy Production Principle* (**MEPP**) have been widely discussed recently. Dewar analyses how these are consistent both with Jaynes' MaxEnt formalism and Onsager's linear transport theory, and elegantly shows how Maximum Entropy Production (**MaxEP**) can quantitatively emerge from within a non-equilibrium statistical mechanical framework (2003 [23], 2005 [24]). Bruers subsequently discusses and amplifies Dewar's treatment (2007 [25]) with a 'partial steady state' MaxEP analysis. The useful mini-review of Martyushev and Seleznev (2014 [26]) also explicitly describes the limitations on the MEPP, thereby demonstrating the invalidity of various "counterexamples" proposed in the literature. Zivieri and Pacini (2018 [27]) use the MEPP in a real biochemical application to living systems.

It is also worth noting that although QGT has a close relationship with the formalism of statistical mechanics (as discussed in PJ21) it is *not* a statistical theory. That is to say, the QGT analysis of this present work is explicitly in the *few-body* context (with small numbers of DoFs) whose results do not intrinsically rely on statistical effects. The geometrical entropy effects described here are therefore *analytic and exact*: with the proviso that such an 'analytical' treatment can necessarily only be performed for a few-body problem; indeed, any larger numbers of degrees of freedom would then require (or imply) a more 'statistical' analysis (with a formalism offered in PJ21 [7]).

*Action* is defined as a path integral of the appropriate kinematic Lagrangian. But PJ19 (their Eq.12) isomorphically defines the **Exertion** as a path integral of the *entropic* Lagrangian. Action is in the energy-and-time or momentum-and-length domains (with units of Planck's constant), where exertion is in the entropic-momentum and hyperbolic

space domain (with units of Boltzmann's constant). Moreover, just as there is a *Principle of Least Action* so there is also (an exactly isomorphic) *Principle of Least Exertion*.

*Exertion* is a somewhat elusive concept since hyperbolic space is different from the conventional Euclidean representation of spacetime (although it obeys the familiar Minkowski metric): Euclidean geometry is convenient, but only a local approximation to the geometry of hyperbolic space. Note particularly that galaxies are enormous and must be treated in hyperbolic space: they are spread across a large canvas of spacetime where local linear (Euclidean) geometries must fail (but noting that both hyperbolic and Euclidean representations of spacetime should be considered flat, according to Penrose [28]).

We have previously constructed the appropriate (non-relativistic) *entropic* Hamiltonian, proving that it satisfies the Euler–Lagrange relations (PJ19 Eq.11 and context). In the present work we derive a fully relativistic entropic Hamiltonian (where the space-like and time-like entropic momentum terms are properly interchangeable), showing also why this is necessary to develop the treatment of *entropy production* needed to analyse the evolution over time of MaxEnt systems far from thermal equilibrium. Idealised spiral galaxies certainly are far from equilibrium (since the central supermassive black hole must grow): QGT methods are probably critical to establishing a full account of such galactic evolution, since QGT treats the logarithmic double-spiral shape as a primary geometrical property (that is, *independent* of any theories of gravity) rather than as an emergent property of gravitationally driven density wave dynamics [29]. Onsager's seminal treatment of thermodynamics has been applied previously to galaxies, as described in the extensive review by Chavanis [30] who points out (citing Onsager's 1949 treatment [31]) "*that the statistical mechanics of two-dimensional vortices and self-gravitating systems present a deep analogy despite the very different physical nature of these systems*". Chavanis and Sommeria [32] have developed this thermodynamical approach for (idealised) elliptical galaxies, in particular looking for Maximum Entropy system responses.

In an extended section we generalise Onsager's original treatment of entropy production [16], presenting a proof of the limit at which our treatment becomes exactly equivalent to his. For a secure understanding of real galaxies we will need a rigorous thermodynamical framework: even if in the present work we have to simplify the treatment for an idealised system, a part of our purpose here is to establish some general results to aid further development.

## 2. Hyperbolic Space in QGT

To treat the logarithmic double-spiral (whose axis is singled out as the $x_3$ direction) we use a hyperbolic spacetime, with a space-like dimension $q = R \ln(x/R)$, where $x$ is a normal (Euclidean) space measure and R is a (Euclidean) normalising metric which can often be associated with a radius of the structure under consideration; and with the *entropic* Lagrangian $L_S(q, q', x_3)$ equations based upon the hyperbolic spacetime dimensions $q$, and their spatial derivatives $q'$ with respect to the $x_3$ axis. The associated *entropic* Hamiltonian $H_S(q, p, x_3)$ is defined (PJ19 [1] Appendix B), as is conventional, using the entropic momentum $p$. Parker and Jeynes (PJ19 [1], Appendix B: Introduction) note that "*in our analysis, the entropic Lagrangian is defined in hyperbolic 3-space $q$, and its variation is performed with respect to the Euclidean $x_3$ spatial parameter (thus, $q' \equiv \partial q/\partial x_3$). This is in contrast to the conventional energetic approach, where the kinematic Lagrangian is defined in Euclidean 3-space $x$, and varied according to the time parameter $t$*". Thus, the kinematic ("*action*") representation has time-like qualities whereas the entropic ("*exertion*") representation is more space-like.

It is also worth noting in this context that $x_0$ ($\equiv ct$) and $x_3$ are conjugate quantities in the Pauli algebra (see PJ19 [1] Appendix A), so that $x_3$ could also be termed 'geometric time', in contrast to the conventional (kinematic) time $t$.

Expanding this a little, PJ19 *derives* the entropic Hamiltonian of the logarithmic double spiral (a Maximum Entropy geometry). Conventionally the Hamiltonian is defined as (assuming the summation convention): $\mathcal{H}(\boldsymbol{p}, \boldsymbol{x}, t) = p_n \dot{x}^n - \mathcal{L}(\boldsymbol{x}, \dot{\boldsymbol{x}}, t)$; where $\mathcal{L}$ is the

Lagrangian, $\{p, x\}$ are the conjugate variables (for example, "momentum" and "position"), and $t$ is time (where $\dot{x} \equiv \partial x / \partial t$). But the *entropic* Hamiltonian is defined for a specific *geometry* with a preferred axis $x_3$, and with the *entropic* conjugate variables $\{p, q\}$ ("entropic momentum" and "hyperbolic position") and the special (preferred) axis $x_3$ replacing *time*. Entropy is space-like where energy is more time-like.

The conjugate variables for the entropic Lagrangian–Hamiltonian analysis are $\{q, p\}$ which are isomorphic to the position–momentum conjugates $\{x, p\}$ of the kinematic Lagrangian (see the list of isomorphisms in PJ19 [1] Table 1). But in the entropic treatment of QGT, $q$ is the hyperbolic position with dimension of length (and units: [m]); and $p$ is entropic momentum with dimension of entropy per unit length (and units: [(J/K)/m] or [J/K·m]). Note that the canonical relations hold as usual for the entropic $\{q, p\}$: that is, the equations of state are given as expected by $q' = \partial Hs / \partial p$, and $p' = -\partial Hs / \partial q$ (see PJ19 [1] Appendix B Eq.B.17), where as before the primes indicate differentiation with respect to the space (not the time) dimension ($\partial / \partial x_3$: thus, the hyperbolic velocity $q'$ is dimensionless). In QGT the time dimension does not appear explicitly (but it is implicit in the 2nd Law). In addition, in a separate treatment PJ21 [7] show how the canonical relations also define an entropic phase space with properties that obey Liouville's theorem and associated expressions based on the Poisson bracket.

In QGT, because $q'$ is now dimensionless (as is required for taking the logarithms used in hyperbolic space) there is an ambiguity between $q'$ and its inverse $1/q'$. The implications of this are also explored by PJ21 (see their Eq.23 and context), but here we merely observe that the entropic momentum is defined as $p \equiv m_S / q'$ (PJ19 Eq.9b), but when $q'$ is apparently greater than unity (that is, equivalent to the *phase* hyperbolic velocity $q'_\varphi$, see PJ21 Eq.23) the inverse case applies, and we must use $q'_\varphi \equiv p / m_S$.

The entropic mass $m_S$ is given by $m_S \equiv i\kappa_0 k_B$ (where $i^2 = -1$ and $\kappa_0$ is a parameter of the system that looks like a "wavenumber"; see PJ19 Eq.9c). That is, the entropic mass $m_S$ is an imaginary quantity, and it scales with Boltzmann's constant $k_B$. The parameter $\kappa_0$ is essentially the system's wavenumber (or the inverse of the "holographic wavelength": see PJ21 for a discussion of the *holographic principle* in this context). It is surprising that something we call "entropic mass" is mathematically an imaginary quantity, but the whole idea of "*info-entropy*" rests on properties of analytical continuation: that is, complex 4-space is central to this whole formalism. In fact, PJ19 (see their Eq.1b) is *explicit* about entropy and information being mutually Hodge duals.

Why use hyperbolic spacetime? The underlying reason is that the laws of thermodynamics are valid at all scales, and it is only in hyperbolic space that the (dimensionless) "velocity" variable $q'$ and its inverse can be considered to be "equivalent" to each other in the sense of yielding entropic properties with reciprocal symmetries (these ideas are developed in PJ21). QGT has been demonstrated at the galactic scale (PJ19), at the molecular scale (for Buckminsterfullerene [33]) and at the nuclear scale [34]. So hyperbolic space clarifies the essentials of systems: both supermassive black holes and alpha particles are represented in QGT as unitary entities requiring only four scalar quantities for a complete specification: the mass, the charge, the spin, and a scaling parameter ($\kappa_0$) which, for black holes, is related to the Planck length (see the discussion of the Bekenstein–Hawking Equation in both PJ19 and PJ21) and for alpha particles is related to the diameter of the proton (see [34]).

The Principle of Relativity has been applied to position (the Universe is assumed to look (essentially) the same from any position), velocity (Special Relativity) and acceleration (General Relativity): if the appropriate space for thermodynamics is hyperbolic (which is essentially scale-less) then a *Relativity of Scale* also applies. That is, thermodynamics applies equally at all scales. Accordingly, QGT has been demonstrated at almost all scales, from the sub-atomic to the galactic.

### 3. The Relativistic Entropic Hamiltonian in QGT

In the same way that we can take the Hamiltonian as expressing the *energy* of the system, given by the sum of the kinetic ($T$) and potential ($V$) *energy* terms ($H = T + V$), we consider the (non-relativistic) *entropic* Hamiltonian $H_S$ to be the sum of the kinetic ($T_S$) and potential ($V_S$) *entropy* terms (PJ19 [1] Appendix B Eq.B.13b, p.30: note that the non-relativistic Lagrangian is given in PJ19 Eq.B.40a, and Eq.B.13b of PJ19 is explicitly evaluated in their Equation (Eq.B.40b)):

$$H_S = T_S + V_S = \sum_{n=1}^{3} -m_S \ln q'_n + V_S(q_n) = \sum_{n=1}^{3} m_S \ln q'_{\varphi,n} + V_S(q_n) \tag{1}$$

where the subscripts $S$ emphasise that the relevant quantities are *entropic* and the axis of the double-spiral (the $x_3$ direction) is not symmetrical with the other two directions ($x_1$ and $x_2$).

As already discussed, the entropic mass $m_S$ is an imaginary quantity, and the conjugate variables of the entropic Hamiltonian are the vectors in hyperbolic 3-space $\{p, q\}$, where the (dimensionless) hyperbolic velocity $q'$ satisfies the canonical relations. That is, $T_S$ is a function of $p$ (and hence $q'$) alone, and $V_S$ is a function of $q$ alone, as required. However, PJ19 made no distinction between the group $q'$ and phase $q'_\varphi$ hyperbolic velocities (see further in PJ21), where $q' \leq 1$ and $q'_\varphi \geq 1$, with $q' = 1/q'_\varphi$. Both hyperbolic velocities yield the same magnitude for the kinetic entropy $T_S$, but of different sign due to its logarithmic character. Noting that $p$ tends to be greater than $m_S$ (as calculated for the double-helix DNA forms in PJ19 and also as implied by the analysis below) we assume that the phase hyperbolic velocity is the quantity to be employed in Equation (1), so that the inverse identity for the group hyperbolic velocity is applicable in this case, $q'_\varphi = p/m_S$. Setting $V_S = 0$ we can rewrite Equation (1) as:

$$H_S = \sum_{n=1}^{3} m_S(\ln p_n/m_S) = \sum_{n=1}^{3} m_S(\ln p_n - \ln m_S) = m_S \ln p - m_S \ln m_S \tag{2}$$

We note that the entropic momentum $p$ is a vector in (hyperbolic) Minkowski 3-space: we omit the basis vectors in this space for clarity, but they are explicit in PJ19 (see their Eq.1 and context, including their Appendix A which also has a careful discussion of the appropriate Clifford algebra).

Setting the entropic potential $V_S$ to zero for Equation (2) just follows the usual simplifications used in Special Relativity. In the case considered by PJ19 (the Milky Way) this simplification is merely formal since they have proved that (in this case) certain approximations are valid, which means that "*the hyperbolic accelerations* [for the logarithmic double spiral] *are therefore all zero indicating the effective absence of any entropic forces or any entropic potentials,* $V_S = 0$" (see PJ19 [1] Appendix B after Eq.B.34c). They go on to say that "*it is clear that in hyperbolic space the entropic Hamiltonian of a logarithmic double spiral . . . is mathematically equivalent to that of a double helix*" (for which see also their Appendix B after Eq.B.43: "*In* [the double-helix] *case, the entropic field reduces to* [a constant] *term . . . that is . . . we can equivalently assume* $V_S = 0$"). Note that in the general case, the full (unapproximated) entropic Hamiltonian $H_S$ for the logarithmic double-spiral is indeed a constant of the system (see PJ19 [1] Appendix B and the last line of their Eq.B.40b).

Equation (2) is manifestly non-relativistic, since the two conserved quantities, the entropic Hamiltonian and the entropic momentum, $H_S$ and $p$, as described by the Euler–Lagrange Equations (and using Noether's Theorem) are clearly not interchangeable. Hence, before we can make any progress, we need a credible entropic Hamiltonian that obeys the conventional rules of relativity.

The conventional Hamiltonian of kinematics gives the total energy of the system (a conserved quantity): the non-relativistic kinematical Hamiltonian $H$ is frequently used in the Schrödinger Equation and given in the non-relativistic approximation (and in the absence of any fields) by $H = p^2/2m$ (where in this case $p$ is the kinematic momentum and

$m$ is the inertial mass as usual). In Special Relativity the total energy (ignoring the potential energy terms) is given by $E_0^2 = c^2 p^2 + m^2 c^4$ ($c$ is the speed of light as usual), and we have:

$$H \equiv E_0 = \left( c^2 p^2 + m^2 c^4 \right)^{1/2} \approx p^2/2m + mc^2 \, (\text{for } p \ll mc) \tag{3}$$

It is also worth pointing out that the additional constant term $mc^2$ (the rest mass energy) is a background term which plays no part in the classical Lagrangian calculations since it simply differentiates away and can be ignored. Moreover, the kinematic momentum $p$ is incommensurate with the inertial mass $m$: they have different units. In kinematics, $c$ is needed to make $m$ commensurate with $p$.

Considering the similar approximation of Equation (2) (that is, ignoring the potential energy terms; that is, setting $V_S = 0$) let us name the *un*approximated relativistic entropic Hamiltonian we seek as "$p_0$"; the subscript "0" indicating that this is a *time-like* entropic momentum. Then, taking the logarithm and form in Equation (2) as suggestive, we suppose:

$$H_S = m_S \ln(1 + p_0/m_S) \tag{4}$$

such that for $p_0 \ll m_S$ (isomorphic to $p \ll mc$ in the kinematic case of Equation (3)) then $H_S \approx p_0$. Thus:

$$m_S \ln \left( 1 + \frac{p_0}{m_S} \right) = m_S \ln p - m_S \ln m_S \tag{5}$$

This can now be straightforwardly rearranged as (remembering that $m_S$ is imaginary):

$$p = p_0 \pm m_S \tag{6}$$

which are clearly the solutions to the quadratic expression

$$p^2 = p_0^2 - m_S^2 \tag{7}$$

where the conserved quantities (the entropic Hamiltonian $p_0$ and the entropic momentum $p$) are now clearly interchangeable. We have a relativistic entropic Hamiltonian! Writing out the 3-vector $\boldsymbol{p}$ in its components we have $p^2 = p_1^2 + p_2^2 + p_3^2$, and

$$p_0^2 = p_1^2 + p_2^2 + p_3^2 + m_S^2 \tag{8}$$

Subscript numbering in Equation (8) conforms to the conventional numbering for the space-time dimensions $\gamma_\mu$ where $\mu = \{0,1,2,3\}$; that is, $\gamma_0$ is the time-like axis and $\gamma_{1,2,3}$ are the space-like axes, using the geometric (Clifford) algebra notation. Note the absence of $c$ in Equation (8). This is because the speed of light $c$ in the kinematic domain is isomorphic to a maximum hyperbolic velocity $q' = 1$ in the entropic domain (recall that $q'$ is dimensionless); moreover, in the entropic domain the entropic momentum is *commensurate* with the entropic mass.

It is clear that just as the kinematic Hamiltonian $H = p^2/2m$ is the non-relativistic approximation to the relativistic energy–momentum expression $E_0^2 = c^2 p^2 + m^2 c^4$, so the entropic Hamiltonian $H_S = m_S \ln p/m_S$ is a non-relativistic approximation to the entropic dispersion relation $p_0^2 = p^2 + m_S^2$. It is also interesting to note that, although they are non-relativistic approximations, both $H$ and $H_S$ still represent conserved quantities (as according to Noether's theorem) in their respective Hamiltonian–Lagrangian equations of state; although the full relativistic conservation laws require that it is $E_0$ and $p_0$ that are conserved in their respective kinematic and entropic domains.

In kinematics, considering the special relativity dispersion relation, the energy (that is, the kinematic Hamiltonian $H$) is equivalent to 'time-like momentum', while the conventional momentum terms are simply the 'space-like momentum' terms. That is why the entropic Hamiltonian $H_S$ is designated as equivalent to $p_0$ since $p$ is an entropic momentum (denoted as time-like by the subscript "0"). The quantity $p_0$ could therefore be considered as

an '*entropic energy*', although it is not yet entirely clear what such an idea implies, physically (except that it must be positive-definite). At present we restrict ourselves to developing the formalism and leave the interpretation of the isomorphism between the kinematic and the entropic to future work.

We should also point out that to speak properly about "relativistic invariance" one should also determine the appropriate Lorentz-like transformations. Fortunately, this has already been done by Parker and Walker, 2010 [35].

## 4. Relationship to Onsager's Differo-Integral

We now show how our relativistic entropic Hamiltonian (the positive solution to Equation (6)) is equivalent to Onsager's celebrated differo-integral (over a volume) for the rate of increase of entropy of a system (Eq.5.11b of [16]):

$$\dot{S} + \dot{S}^* - \Phi(J, J) = \int \left[ J_n \frac{\partial}{\partial x^n} \left( \frac{1}{T} \right) - \phi(J, J) \right] dV \tag{9}$$

where Einstein's summation convention is in use.

The quantity $\dot{S}$ is known as the *entropy production* (that is, the rate of increase of entropy), and the quantity $\dot{S}^*$ is the entropy given off to the surroundings. In Equation (9) the quantity $J_n$ is a heat flux or heat flow term (in the three space directions), $T$ is the temperature, and $V$ is the volume. The function $\phi(J, J)$ (and its volume integration $\Phi$) is known as the "*dissipation function*", and according to [16] is interpreted as a "*potential*" function for the "*mutual interaction of frictional forces*", where such forces are dissipative (entropy producing). Thus, $\phi$ is positive definite (in accordance with the 2nd Law). Onsager employs the variational principle to demonstrate that the quantity $\dot{S} + \dot{S}^* - \Phi(J, J)$ is a *maximum* for any system, such that one can write:

$$\delta \left[ \dot{S} + \dot{S}^* - \Phi(J, J) \right] = \delta \int \left[ J_n \frac{\partial}{\partial x^n} \left( \frac{1}{T} \right) - \phi(J, J) \right] dV = 0 \tag{10}$$

again, using the summation convention.

Equations (9) and (10) can be simplified if the system under consideration is isolated, such that no heat flows across its boundary; that is, we can assume $\dot{S}^* = 0$. It is noteworthy that Onsager wrote his paper in 1931, long before Shannon expounded his probabilistic approach to entropy in his famous 1948 paper [2]; we have essentially used the Shannon definition of entropy to derive the variational equation for our entropic Hamiltonian $H_S$ (PJ19 [1], Table 1):

$$\delta S = \delta \int H_S dx_3 = 0 \tag{11}$$

A key difference between Equations (10) and (11) is that whereas Equation (10) is a volume integral, Equation (11) is a line integral. This is because Equation (11) assumes a holomorphism requiring an axis of rotational symmetry in the $\gamma_3$ direction. Thus, we first need to invoke a suitable cross-sectional area $A$, to make Equations (10) and (11) mutually commensurate. $A$ is constant with respect to $\gamma_3$ since in hyperbolic space the logarithmic double-spiral behaves as a double-helix (see the context of Equation (2)).

Thus, we assume the heat flux $J_3$ is across a cross-sectional area $A$ in the $\gamma_3$ direction, and for our entropic geometries of interest (in this case the 'cylindrical' geometry of the double helix) we assume that $J_1 = J_2 = 0$. The double helix geometry also implies $\partial/\partial x_1 = \partial/\partial x_2 = 0$. In addition, Onsager's equation (Equation (10)) is intrinsically based upon temporal derivatives of entropy (and energy), which are not present in Equation (11).

We therefore redeploy the relativistic entropic momentum term $p_0$ in place of the original (non-relativistic) entropic Hamiltonian $H_S$:

$$\delta S = \delta \int H_S dx_3 = \delta \int p_0 dx_3$$
$$\Rightarrow \delta \dot{S} = \delta \left( \frac{\partial S}{\partial t} \right) = \delta \left( \frac{\partial}{\partial t} \int p_0 dx_3 \right) = \delta \left( \int p_0 d \frac{\partial x_3}{\partial t} \right) = \delta (\int p_0 dc) = \delta(cp_0) \tag{12}$$



In Equation (12) we have assumed that $p_0$ is time-independent; that is, we assume that the entropic momenta (both time-like as well as space-like) of the stable spatial geometries of interest do not change over time (that is, $\delta \dot{S} = 0$). This is true for our maximum entropy systems, and the same is true in conventional kinematics (without dissipation and without the presence of potential fields) where the total energy $E_0$ is a constant of the system. We have also assumed that the velocity quantity $\partial x_3 / \partial t = c$ is simply the speed of light, which is also a relativistically invariant universal constant.

Equation (12) shows that the time-like entropic momentum term $p_0$ is equivalent to the entropy production $\dot{S}$ when made commensurate by the normalising constant $c$. This is important for the physical interpretation of $p_0$, thus:

$$\dot{S} \equiv c p_0 \equiv c H_S \tag{13}$$

Since the Hamiltonian can be offset by a constant factor, we can rewrite the first line of Equation (12) with a (positive-definite) offset $\Phi/c$, where we can interpret $\Phi$ as the volume-integrated dissipation function of Equation (9):

$$
\begin{aligned}
& \delta S = \delta \int (p_0 + \Phi/c)\,\mathrm{d}x_3 \\
\Rightarrow\quad & \delta \dot{S} = \delta(c p_0 + \Phi) \\
\Rightarrow\quad & \dot{S} - \Phi = c p_0
\end{aligned}
\tag{14}
$$

To prove complete consistency with Onsager's variational differo-integral of Equation (10) we now employ the positive solution of Equation (6), $p \equiv p_n \gamma^n = p_0 + m_S$ (summation convention: as before, $\gamma^n$ is the set of basis vectors describing the three spatial coordinates of Minkowski 4-space—see PJ19 Eq.1), which is the key expression linking Onsager's equation (Equation (9)) to our geometric entropy analysis. We substitute in Equations (12) and (14):

$$
\begin{aligned}
\delta\left(\dot{S} - \Phi\right) &= \delta \frac{\partial}{\partial t} \int p_0\,\mathrm{d}x_3 = \delta \frac{\partial}{\partial t} \int (p_n \gamma^n - m_S)\,\mathrm{d}x_3 \\
&= \delta \int \frac{1}{A} \frac{\partial}{\partial t}(p_n \gamma^n - m_S) A\,\mathrm{d}x_3 \\
&= \delta \int \frac{1}{A} \frac{\partial}{\partial t}\left(m_S \frac{\partial x_3}{\partial q_n} \gamma^n - m_S\right)\mathrm{d}V
\end{aligned}
\tag{15}
$$

(summation convention). In Equation (15) we have introduced a notional cross-sectional area $A$, so that the infinitesimal volume is given by $\mathrm{d}V = A\,\mathrm{d}x_3$, and where we also use the relation for the entropic momentum terms (PJ19 Eq.9b): $p_n = m_s(\partial q_n / \partial x_3)^{-1} = m_S / q'_n$.

We emphasise that Onsager's analysis is based on a temperature $T$, whereas our analysis is not temperature-based and is resolutely purely *entropic* in nature; no notion of "temperature" is ever employed in our entropic analysis. This is because temperature is the coupling coefficient linking energy and entropy ($T = \mathrm{d}E/\mathrm{d}S$) enabling translation between the "entropic" and "energetic" domains. The isomorphism between the entropic and kinematic descriptions is illustrated in some detail in PJ19 [1] Table 1.

To demonstrate the equivalence of our approach to Onsager's, we insert a nominal (spatially uniform in the $x_1$–$x_2$ directions) temperature term $T$ into Equation (15). We also explicitly expand the entropic mass term $m_S$ ($m_S \equiv i \kappa k_B$), also forcing it to be positive-definite as is required for a dissipation function term. We also take advantage of the (Fourier) identity relationship $\mathrm{d}/\mathrm{d}x_3 \equiv i\kappa$ (PJ19 Eq.15) so that we can write:

$$
\begin{aligned}
\delta\left(\dot{S} - \Phi\right) &= \delta \int \frac{1}{A} \frac{\partial}{\partial t}\left(m_S T \frac{\partial (x_3/T)}{\partial q_n} \gamma^n - |m_S|\right)\mathrm{d}V \\
&= \delta \int \frac{\partial}{\partial t}\left(\frac{i\kappa k_B T}{A} \frac{\partial (x_3/T)}{\partial q_n} \gamma^n - \frac{\kappa k_B}{A}\right)\mathrm{d}V \\
&= \delta \int \frac{\partial}{\partial t}\left(\frac{k_B T}{A} \frac{\partial}{\partial x_3} \frac{x_3 \partial (1/T)}{\partial q_n} \gamma^n - \frac{\kappa k_B}{A}\right)\mathrm{d}V \\
&= \delta \int \left(\frac{\partial}{\partial t}\left(\frac{k_B T}{A}\right) \frac{\partial (1/T)}{\partial q_n} \gamma^n - \frac{\partial}{\partial t}\left(\frac{\kappa k_B}{A}\right)\right)\mathrm{d}V
\end{aligned}
\tag{16}
$$

(with the summation convention). We note that the resulting $k_B T$ quantity is clearly an energy term, with the resulting quantity $\partial/\partial t(k_B T/A)$ therefore representing an energy flux term. This allows us to identify the following equivalent relations between our entropic geometry and Onsager's entropy equation:

$$J_3 \equiv \frac{\partial(k_B T/A)}{\partial t} \tag{17}$$

$$\frac{\partial}{\partial x_n}\left(\frac{1}{T}\right) \equiv \frac{\partial}{\partial q_n}\left(\frac{1}{T}\right) \tag{18}$$

$$\phi(J,J) \equiv \frac{\partial(\kappa k_B/A)}{\partial t} = \sqrt{J_3^2}\frac{\kappa}{T} \tag{19}$$

Equation (18) makes clear the distinction between Euclidean and hyperbolic geometries: Onsager's formalism employs a Euclidean geometry where we employ a hyperbolic one. As we have already mentioned (and as discussed by PJ19—see their Eq.9a and context), when the entropic system is being considered close to its holographic boundary, we find that hyperbolic position $q$ closely approximates its Euclidian counterpart $x$. Thus, at least for the particular entropic geometries that we are interested in, where $\partial/\partial x_1 = \partial/\partial x_2 = 0$ and $J_1 = J_2 = 0$, we find that our relativistic entropic Hamiltonian formalism is therefore exactly equivalent to Onsager's variational approach. It is worth noting that the dissipation functions in Equations (9) and (14) are closely identified with the 'entropic mass' $m_S$ term that we have previously defined; that is, the term $\phi$ can be understood to be the entropic mass-density flow, whereas $\Phi$ is the volume-integrated entropic mass-density flow. Being mass-like in character, it is also clear that all of $m_S$, $\phi$ and $\Phi$ must therefore also always be positive-definite.

Whereas Onsager considers the entropy production $\dot{S}$ and $\Phi$ to be at a maximal extremum (in accordance with the 2nd Law), we have considered the entropy $S$ in our variational analysis and have proved that it is also at a maximal extremum (MaxEnt) for our geometric structures of interest. Our analysis has also identified the quantity $p_0$ (the time-like entropic momentum which could be considered analogous to an 'entropic energy'), which is equivalent to the entropic Hamiltonian, and which we now find is also directly proportional to the entropy production $\dot{S}$ according to Equation (13) and is therefore also positive-definite in accord with the 2nd Law. We noted above that energy is itself a positive-definite quantity: its entropic isomorph $p_0$ is also similarly constrained through the application of the 2nd Law—this is an intriguing relationship between energy's positive-definite nature and the 2nd Law which we expect future work to illuminate.

Therefore, it is now clear that we can therefore also generalise Onsager's differo-integral for the hyperbolic-space ($q$ geometry) case (with the summation convention):

$$\dot{S} + \dot{S}^* - \Phi(J,J) = \int\left[J_n\frac{\partial}{\partial q^n}\left(\frac{1}{T}\right) - \phi(J,J)\right]\mathrm{d}V_q \tag{20}$$

where the volume $V_q$ is assumed to be in the appropriate hyperbolic space.

## 5. Conservation of Entropy Production

The entropy production $\dot{S}$ is a conserved quantity (according to Noether's theorem and the physics of Special Relativity) since the entropic Euler–Lagrange Equations are satisfied and given by (PJ19, Eq.13b):

$$\frac{\mathrm{d}}{\mathrm{d}x_3}\frac{\partial L_S}{\partial q'_n} - \frac{\partial L_S}{\partial q_n} = 0 \qquad (n \in \{1,2,3\}) \tag{21}$$

where the entropic Lagrangian $L_S$ is related to the entropic Hamiltonian via the Legendre transformation (PJ19, Eq.11) $L_S = 3m_S - p_0$.

Equation (21) indicates that the space-like entropic momentum components $p_n \equiv \partial L_S/\partial q_n$ (PJ19 [1], Appendix B, Eq.B.16c) must be conserved as required by Noether's theorem. Taken with the relativistic entropic momentum dispersion relation (Equation (8)) it is reasonable to assume that the entropic Hamiltonian $p_0$ is also conserved along with the entropic mass $m_S$—using the same mathematical and physical reasoning that makes energy, kinematic momentum and inertial mass conserved quantities in Special Relativity. The fact that the entropy production is simply the product of the entropic Hamiltonian with the speed of light $c$ (a universal constant) means that the entropy production $\dot{S} \equiv cp_0$ must have equivalent properties to the entropic Hamiltonian, and consequently should also be a conserved (constant) feature of any entropic system under consideration (provided the usual conditions hold, such as no external potential fields).

## 6. The Entropy Production of an Idealised Spiral Galaxy

Here (as in PJ19) we have considered a zeroth-order model of an isolated logarithmic double-spiral galaxy, consisting of a pair of spiral arms with a supermassive black hole at its centre. Such a geometry has maximum entropy and is therefore stable; but is not thereby necessarily in static equilibrium. In fact, since black holes necessarily grow, the galactic entropy must be increasing. This is a boundary condition: the central black hole in the Milky Way has a Hawking temperature of 15 fK where the cosmic microwave background (**CMB**) temperature is about 3 K: photons at least must flow into the black hole.

We are here considering a simpler case: our model galaxy is isolated (we do not consider the CMB). We wish to explore the properties of this highly simplified model. In particular, we will demonstrate the remarkable fact that the entropy production for the isolated logarithmic double-spiral geometry *does not vanish*.

Of course, real supermassive black holes in the centres of real galaxies vigorously accrete hadronic matter in extraordinarily energetic processes, processes that are currently the subject of intense interest [36]. Undoubtedly QGT will provide valuable insights into the real situation, but at present we must start with our zeroth-order model.

Combining Equation (6) (the positive solution) and Equation (14) we consider the entropy production of the galaxy to consist of two parts on the RHS:

$$\dot{S} - \Phi \equiv cp_0 = cp - cm_S \tag{22}$$

We will show (Equations (27) and (28)) that the first component, $cp$ (that is, the entropy production due to the entropic momentum $p$) is closely related to the Hawking radiation. It is the second component (related to the entropic mass $m_S$ of the galactic structure) that is the dominant part of the entropy production.

We first consider the entropic momentum component aspect $cp$, using the basic equation (PJ19 [1], Appendix B, Eq.B.7) for the entropic momentum in the vicinity of the black hole event horizon of Schwarzschild radius ($r_{BH}$) where the radius $R$ of the local structure is given by $R = r_{BH}$:

$$p = \frac{k_B}{r_{BH}} \tag{23}$$

The black hole Schwarzschild radius for a black hole of mass $M_{BH}$ is given by:

$$r_{BH} = \frac{2GM_{BH}}{c^2} \tag{24}$$

where $G$ is the gravitational constant. We can then write the associated entropy production:

$$\dot{S} = cp = \frac{ck_B}{r_{BH}} = \frac{c^3 k_B}{2GM_{BH}} \tag{25}$$

Multiplying by the black hole temperature $T_{BH}$, which is given by:

$$T_{BH} = \frac{\hbar c^3}{8\pi G M_{BH} k_B} \tag{26}$$

(where $\hbar$ is the reduced Planck constant as usual) we then find that the associated power [J/s] associated with that entropy production is given by:

$$P = cp T_{BH} = \frac{c^3 k_B}{2 G M_{BH}} \frac{\hbar c^3}{8\pi G M_{BH} k_B} = \frac{\hbar c^6}{16\pi G^2 M_{BH}^2} \tag{27}$$

which is similar to the Hawking radiation, given by [36]:

$$P_{Hawking} = \frac{\hbar c^6}{1530\pi G^2 M_{BH}^2} \tag{28}$$

Using the parameters of the Milky Way ($M_{BH} = 4.3 \times 10^6$ M$_\odot$) the Hawking radiation of the central supermassive black hole of the Milky Way is equivalent to a (negative) entropy production of $3.3 \times 10^{-28}$ J/K·s (Equation (28)), and the entropic momentum component of the entropy production of the idealised Milky Way galaxy is similar: $3.2 \times 10^{-25}$ J/K·s (Equation (27)).

Next, we consider the component of the overall galactic entropy production due to the intrinsic entropic mass $m_S$ of the galactic structure. The entropic mass at the black hole's Schwarzschild radius is given by (ref. [1]):

$$m_S \equiv i\kappa_{BH} k_B = i\frac{2\pi}{l_P} k_B \tag{29}$$

where $l_P$ is the Planck length given by:

$$l_P = \sqrt{\frac{\hbar G}{c^3}} \tag{30}$$

Since entropy production is a positive-definite quantity (as required by Equation (16)) we can write the second component of the galactic entropy production as:

$$\Phi = c|m_S| = \frac{2\pi}{l_P} c k_B = 2\pi\sqrt{\frac{c^5}{\hbar G}} k_B \tag{31}$$

For the Milky Way in the vicinity of its supermassive black hole, and with $l_P = 1.616 \times 10^{-35}$ m, and $c = 3 \times 10^8$ m/s, we obtain the numerical value for the entropy production (dissipation function) of the Milky Way: $1.6 \times 10^{21}$ J/K·s. That is, the *entropic mass* component of the entropy production of the idealised Milky Way galaxy is 46 orders of magnitude larger than the *entropic momentum* component ($3.2 \times 10^{-25}$ J/K·s, Equation (27); comparable to the Hawking radiation, Equation (28)).

## 7. Discussion

Entropy production in galaxies is key to galactic evolution, since it is a conserved quantity in the idealised representation: even in an evolving MaxEnt system it must be a constant. That is, to a very good approximation it specifies how fast the supermassive black hole at the galactic centre is growing—and this quantity is a *constant* (in the absence of any additional entropic potential fields)! It turns out that ideal galaxies are "simple" systems because the logarithmic double-spiral can be treated analytically, and because galaxies can be treated quite well as isolated systems—which sunflowers and cyclones certainly are not!

The present treatment is of an *idealised* spiral galaxy: that is, the energy (and entropy) flow from the CMB is neglected since the galaxy is considered as isolated from the environment. In addition, the effects of stellar material are ignored: this is a reasonable

"zeroth-order" approximation since it is well-known that so-called "dark matter" (and not the sensible hadronic matter) dominates the virial mass of galaxies. QGT is new, unfamiliar, and consequently conceptually rather challenging. We regard the present analysis as an essential first step in a realistic treatment of the thermodynamics of spiral galaxies, even though the calculated entropy production $\Phi$ for the (idealised) "Milky Way" is apparently too small.

Of course, in real galaxies the central supermassive black hole grows essentially by highly energetic accretion processes (such as Eddington or so-called super-Eddington accretion [37]), which may involve entropy production orders of magnitude larger again than the rates we have calculated for the idealised galaxy; proper consideration of other relevant physical phenomena and parameters (accretion disk temperatures, stellar interactions and the effects of inter-stellar gas, gravitational density waves etc.) might account for the additional entropy production over and above the quantity predicted by our zeroth-order model.

We point out that relativistic jets (in which there has been intense interest for decades: see [38] for a recent relevant example) may have a significance in this context. Spiral galaxies (on a QGT reading) all have central supermassive black holes (BHs), and therefore necessarily increase their entropy (since black holes necessarily grow). In QGT, accretion discs in the galactic plane (increasing the BH entropy) are properties of the stellar mass geometry in the vicinity of the black hole, but the axial relativistic jets (reducing the BH entropy) are a galactic property that ensures the conservation of entropy production. The jets (on a QGT reading) are phenomena conforming to the PLE geometry with a non-local mechanism that enables the proper balance of positive and negative components of galactic entropy production, as per Onsager's equation (Equation (9)) and our Equation (22).

It is also worth commenting upon the *sign* of the component of the entropy production (Equation (27)) that is comparable to the Hawking radiation even though it is almost insignificant when compared to the magnitude of the larger component (Equation (31)). Clearly, the Hawking radiation represents mass (and therefore entropy) *loss* in contrast to the larger component associated with the accretion of mass by the black hole (and a positive entropy production for it). In our analysis, although $\dot{S}$ is assumed positive definite, yet even Onsager's equation (Equation (9)) allows for a negative sign in front of the dissipation function $\Phi$, which we showed is closely related to our entropic mass term $m_S$ (Equations (16) and (19)). Indeed, it is also noteworthy that in the context of minimum entropy production Ilya Prigogine suggests that " ... [entropy] *production expresses a kind of 'inertial' property of nonequilibrium systems*" [39], thus alluding to the fact that there is a 'mass-like' aspect to entropy production, which we see here being expressed by the entropic mass $m_S$. Indeed, Equation (6) indicates that the two commensurate components (the entropic momentum $p$, and the entropic mass $m_S$) might intrinsically exhibit different signs and therefore physical behaviours. On the one hand, this is because Equation (6) admits both positive and negative (conjugate) solutions for the entropic mass component $m_S$; while on the other hand $m_S$ is explicitly imaginary with respect to $p$. Thus, this indicates an intrinsically different origin and nature to these entropic phenomena: respectively, more time-like ($m_S$) and more space-like ($p$) in their origin and behaviours. From this perspective, it is perhaps reasonable to assign the time-like ($cm_S$) entropy production term to the "dissipation function" $\Phi$, and the space-like ($cp$) entropy production term to $\dot{S}$ of Equations (9) and (20)—where of course the signs of these terms must be consistent with the frame of reference: either that of the surrounding galaxy or that of the black hole (beyond the event horizon). That is, either side of the hyperbolic boundary, entropy production terms must change sign.

Notwithstanding the unconditional validity of the 2nd Law of Thermodynamics which is intrinsically embedded within the geometric (Clifford) algebra of QGT, in this paper we have only studied the description of QGT within the realm of Special Relativity where the metric of spacetime is flat. Clearly, where the metric is *not* flat (that is, in the presence of gravitational effects, or where the system is accelerating) then the specific formalism

of General Relativity must apply. The application of GR principles to QGT has not been studied in this paper, which therefore represents an important limitation to the validity of QGT theory as currently described. Future work in the development of QGT will be to understand how the current formalism of QGT must be adapted to make it also valid for application within the realm of General Relativity.

## 8. Conclusions

Using QGT we have for the first time calculated the entropy production of an idealised spiral galaxy which is defined only by its central supermassive black hole (that is, it is isolated from the Universe and ignores the presence of stellar material).

We have extended QGT to generalise Onsager's differo-integral formalism for the entropy production of systems in hyperbolic (Minkowski) 4-space: this shows that the relativistic entropic Hamiltonian directly determines the entropy production of the idealised galaxy; and also that both the entropic Hamiltonian and the entropy production are conserved quantities, in the absence of any entropic fields or forces. Applying this to a spiral galaxy (representing a MaxEnt unitary structure) we have demonstrated that the galactic rate of entropy increase remains constant even as the galaxy evolves. We find that the galactic entropy production consists of two components: an extremely small aspect that is related to the Hawking radiation of the central (supermassive) black hole, and a much larger contribution due to the entropic mass of the galaxy.

Extending the analysis to non-isolated systems (such as living entities, as well as weather-like natural phenomena) suggests the existence of a new system invariant (the entropy production) based on the principle of least exertion, that offers a new means to analyse and understand the system under consideration that is complementary to the conventional kinematical analyses based on the principle of least action. The integration of QGT into the overall general relativistic setting also represents an important area of future work.

Such a new QGT-based methodology to understand entropy-driven phenomena (such as black holes, galactic evolution, plant and animal life forms, meteorological manifestations and dynamic chemical equilibria, to name but a few) offers both a powerful new analytical tool to obtain new insights into their operation as well as the possibility for new methods to engineer their performance and efficiency.

**Author Contributions:** Conceptualization, M.C.P. and C.J.; methodology, M.C.P. and C.J.; formal analysis, M.C.P.; writing—original draft preparation, M.C.P.; writing—review and editing, M.C.P. and C.J. All authors have read and agreed to the published version of the manuscript.

**Funding:** This research received no external funding.

**Conflicts of Interest:** The authors declare no conflict of interest.

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
