# Peer review of "A Relativistic Entropic Hamiltonian–Lagrangian Approach to the Entropy Production of Spiral Galaxies in Hyperbolic Spacetime"

_universe, doi:10.3390/universe7090325_

Round 1
Reviewer 1 Report
The authors analyze the entropic aspects of idealized double-spiral galaxies, in particular the Milk Way, on the base of two their previous works (ref.s 1 and 7). More specifically a generalization of the Onsager conditions for the entropy production is proposed and it is shown that the latter can be decomposed into two parts, one comparable to the Hawking radiation, the other ascribable to the accretion disks of real supermassive black-holes.
The results are interesting, mainly because put in evidence the role that the concept of entropy can have in a geometric context, and not only in a standard thermodynamic context.
However, the paper presents several points that must be addressed. First of all the pdf file looks corrupted: several formulas are not readable, e.g. eq.s 1a, 1b, 4, 5b, 6, 9, 17 and many others.
Apparently, the authors have not checked the submitted version. Therefore, in order to perform the review, it is mandatory to provide an amended version of the manuscript. I have tried to interpret the anomalous symbols but of course publication requires a clear version.
As second point, several questions are not discussed but the reader is simply refereed to the previous works of the authors. This makes section 2 rather “obscure”. What is the rationale for several definition the authors introduce, such as entropic mass, velocity and momentum? Is the choice of the entropic Hamiltonian based on some physical consideration?
Reviewer 2 Report
This is a very nice paper dealing with the entropy production of spiral galaxies using the Quantitative Geometrical Thermodynamics (QGT) method within a Lagrangian-Hamiltonian relativistic framework. The authors generalize Onsager’s differo-integral in hyperbolic four-dimensional space. The key result is the calculation of the entropy production of the idealized spiral galaxy. It is shown that the relativistic entropic Hamiltonian determines the entropy production of the idealized galaxy. The formalism is mainly based on that of two recent articles published by the same authorswhose formalism and main results of that paper are often recalled.
I particularly like the formulation of the principle of least exertion as the space-like analogous of the principle of least action and the demonstration of the equivalence of Onsager’s differo-integral principle formulated for the rate of entropy in the Euclidean space and the same principle in the hyperbolic spacetime.
Without any reservation, I recommend this paper for the publication in the Universe journal.
However, prior to the publication, I would suggest to address the following comments/questions:
1) Maybe the title could be shortened writing “… to the Entropy Production …” explaining in the Abstract that Entropy Production corresponds to the Intrinsic Dissipation Characteristics.
2) I appreciate the explanations shown about the choice of a hyperbolic space in place of an Euclidean space by means of the concept of exertion as equivalent to that of action in terms of a path-integral of the entropic Lagrangian and the demonstration of the equivalence with Onsager’s principle but this choice is still not clear to me from a physical point of view. I would suggest a brief and further explanation of this analogy.
3) What is the physical meaning of entropic energy and what is the need to introduce this concept? I think that this concept is certainly of great interest in every field of physics and a further clarification of this concept could be really useful for a reader.
4) The minimum and maximum entropy production principles have been debated so far during the last years and it has been shown that they do not contradict each other. Prigogine’s energy minimum dissipation principle (or theorem) within nonequilibrium thermodynamics and in the framework of local equilibrium and local entropy production was recently applied also to biological systems close to equilibrium but not considering the deviation from equilibrium as in the Onsager’s principle. I would suggest to briefly discuss, in the Introduction, the following papers of the last two decades dealing with maximum and minimum entropy production principles that cannot be ignored:
- Martyushev, L. M. The maximum entropy production principle: two basic questions. Phil. Trans. R. Soc. B 2010, 365, 1333.
- Dewar, R. C. Information theory explanation of the fluctuation theorem, maximum entropy production and self-organized criticality in non-equilibrium stationary states. J. Phys. A: Math. Gen. 2003, 36, 631–641.
- Dewar, R. C. Maximum entropy production and the fluctuation theorem. J. Phys. A: Math. Gen. 2005, 38, L371–L381.
- Zivieri R. and Pacini N. Entropy Density Acceleration and Minimum Dissipation Principle: Correlation with Heat and Matter Transfer in Glucose Catabolism, Entropy 2018, 20, 929.
- Bruers, S. A discussion on maximum entropy production and information theory. J. Phys. A: Math. Theor. 2007, 40, 7441–7450.
5) Does this approach apply only to spiral galaxies or can it be extended to other types of galaxies with similar symmetries?
6) While is this relativistically-invariant description of systems dynamics in the entropic domain isomorphic to the application of special relativity in the energy domain?
Minor: there are some typo errors in a few equations that should be corrected, I suppose that are due to the PDF conversion of the manuscript.
Reviewer 3 Report
see enclosed comments

Round 2
Reviewer 1 Report
I am satisfied with the new version of the paper. I recommend publication.
Reviewer 3 Report
see comments

Author Response
Please see the attachment

This manuscript is a resubmission of an earlier submission. The following is a list of the peer review reports and author responses from that submission.